# Mechanical and Tribological Properties of Polytetrafluoroethylene Modified with Combined Fillers: Carbon Fibers, Zirconium Dioxide, Silicon Dioxide and Boron Nitride

**DOI:** 10.3390/polym15020313

**Published:** 2023-01-07

**Authors:** Andrey P. Vasilev, Nadezhda N. Lazareva, Tatyana S. Struchkova, Aitalina A. Okhlopkova, Sakhayana N. Danilova

**Affiliations:** Federal State Autonomous Educational Institution of Higher Education “M. K. Ammosov North-Eastern Federal University”, 58 Belinskiy str, 677000 Yakutsk, Russia

**Keywords:** polytetrafluoroethylene, carbon fiber, zirconium oxide, silicon oxide, boron nitride, mechanical properties, wear resistance, coefficient of friction

## Abstract

The introduction of combined fillers can effectively improve the mechanical and tribological properties of polytetrafluoroethylene (PTFE). In this work, three different types of nanosized fillers (zirconium dioxide, silicon dioxide, and boron nitride) were introduced in a carbon fiber-reinforced polymer matrix for the development of polymer composite materials (PCM). Tensile and compressive testing were carried out, and the hardness of created PCM was evaluated. It is shown that the compressive strength of PCM increased by 30–70%, and the hardness, increased by 38–55% compared to the initial PTFE. The tribological properties of the developed PCM were evaluated under dry friction conditions. An analysis of the results of an experimental study of wear confirmed that the inclusion of combined fillers (two- and three-component) in PTFE significantly increased wear resistance compared to the polymer matrix with a slight increase in the coefficient of friction. It has been shown that the introduction of three-component fillers has an antagonistic effect on the wear resistance of PCMs compared to two-component fillers. The thermodynamic properties of the composites were analyzed by differential scanning calorimetry and a thermomechanical analyzer. The surface morphology of polymer composites after wear testing was studied by IR spectroscopy and scanning electron microscopy to investigate and suggest a possible mechanism for increasing the wear resistance of the developed composites.

## 1. Introduction

Polymer composite materials are widely used in many industries. Recently, polymer composites with hybrid fillers are becoming increasingly significant due to their multifunctional effect [1,2,3]. Such materials can meet the highest requirements for both mechanical and tribological properties. As parts of friction units, self-lubricating polymer composites are preferred due to their working capacity under dry friction conditions and resistance in aggressive environment [4,5].

At present, in the Arctic region components of equipment and machines with interacting mating surfaces in relative motion, such as bearings, seals or valves, and tribosystems are especially important. For such tribosystems, it is necessary to take into account not only the temperature factor, but also the aggressive environment. Polytetrafluoroethylene (PTFE) is one of the few polymers that can meet such requirements due to its excellent antifriction properties, chemical inertness, frost and heat resistance [6,7,8]. However, the extremely low wear resistance limits its use in friction units, which is due to the molecular and microstructure of the polymer [9]. An effective way to improve the PTFE hardness and wear resistance is to input solid fillers. Carbon materials such as graphite, graphene, carbon nanotubes, and carbon fiber are widely used as PTFE fillers [10,11,12,13]. For the development of tribological materials, carbon fibers are most preferred due to their chemical inertness, high strength, and elastic modulus. A number of works have shown that composites with carbon fibers exceed the glass fibers in terms of strength and tribological characteristics [9,14,15]. The wear mechanism that determines the specific wear rate of polymer composites with short fibers consists of the following processes: the matrix wear and thinning of fiber, fiber fracture, interfacial debonding of matrix/fiber, and fiber removal [16,17]. It is assumed that the mechanism of cracking and delamination occurs sequentially, so they can be considered as a combined process of fiber cracking and separation at the fiber/matrix interface. Wear of polymer composites based on PTFE with carbon fibers (PTFE/CF) at their low content from 1 to 10 wt.% occurs by a similar mechanism. Therefore, in order to increase the wear resistance of PTFE-based composites filled with fibers, dispersed fillers (oxides, layered silicates, sulfides, and graphite) are additionally added into them [18,19,20,21]. In the work of Ronghao L. et al. [22] investigated the effect of plasma-treated CNTs on the mechanical properties of PTFE/CF composites. It is shown that the introduction of combined fillers into PTFE leads to an overall improvement in mechanical properties by 8% relative to composites with a monomodifier. Song F. et al. [23] studied the tribological properties and PV factor of PCM based on PTFE filled with short carbon fibers and fiberglass, as well as MoS_2_. The authors suggested that the improvement in the tribological properties of PCM is due to the fact that the plastic particles of molybdenum disulfide reduce the contact between the glass fiber and the counterbody, which leads to the formation of a more stable transfer film on the friction surface. Zhang et al. studied composite materials based on PTFE filled with carbon and aramid fibers, as well as aluminum oxide. It is shown that the introduction of aramid fibers with aluminum oxide at low contents leads to an improvement in mechanical and tribological properties [24]. At the same time, other types of nanosized fillers remain poorly understood, such as zirconium dioxide (ZrO_2_) and silicon dioxide (SiO_2_) when they were introduced into PTFE/CF, as well as three-component systems, for example, including layered particles in addition to CF and oxide. As a layered particle the hexagonal boron nitride (h-BN) has high thermal conductivity and antifriction properties, it is introduced into polymers to improve their thermophysical and tribological properties [25,26]. Zirconium dioxide and silicon dioxide have attracted increasing interest among nanosized ceramic fillers in recent years. Modification of polymers with zirconium dioxide and silicon dioxide leads to the production of polymer composites with improved tribological and mechanical properties [27,28,29].

The aim of this work is to compare the mechanical and tribological properties of two- and three-component polymer PTFE/CF composites containing nanosized oxides (ZrO_2_, SiO_2_) and boron nitride.

## 2. Materials and Methods

The polymer matrix was presented by PTFE powders (with a particle size of 46–135 µm and density of 2.16 g/cm^3^), trademark PN90 purchased by GaloPolymer, Moscow, Russia. Carbon fibers (R&G Faserverbundwerkstoffe GmbH, Germany) with average length of 0.2 mm, diameter 7 µm, density 1.7 g/cm^3^. Zirconium dioxide ZrO_2_ (Plasmoterm LLC, Russia) is a powder with a particle size of 40–75 nm, with a product purity of 99.75% (irregular shape). Silicon dioxide SiO_2_ (IP Khisamutdinov R.A., Russia) is a powder with a particle size of 20 nm, with a product purity of 99.8% (spherical shape). Hexagonal boron nitride h-BN (Simpleks, Russia) with an average particle size of 100 nm.

Ultrasonic treatment of nanoparticles and mechanical activation of CF were used for the effective dispersion of fillers and stability of the performance of the composite material. Ultrasonic treatment is one of the effective methods for disaggregation of nanosized particles [30,31,32]. In this work, ultrasonic treatment of ZrO_2_, SiO_2_ and h-BN was occured in IL100-6/4 ultrasonic disperser (IN-LAB-Ultrazvuk, Russia) within 5 min. Mechanical activation of the fibers was carried out in Activator-2S planetary mill (Activator, Novosibirsk, Russia) within 5 min at 1500 rpm. Mechanical activation time based on literature data [32,33] Subsequently, composites were obtained in the following way: mixing of dry components in a high-speed mixer, pressing in a hydraulic press at 50 MPa with a holding time of 2 min, sintering in a programmable furnace SNOL 15/900 (Narkūnai (Utena), Lithuania) at 375 °C. Sample preparation of PCM is schematically shown in Figure 1.

Composites based on PTFE were prepared, the detailed compositions of which are listed in Table 1.

The mechanical characteristics of PTFE and PCM were determined according to ISO 527-2:2012 on an Autograph AGS-J universal testing machine (Shimadzu, Kyoto, Japan). The compression stress at the set relative strain (5 and 10%) was determined according to ISO 604:2002 using the same instrument at a strain rate of 1 mm/min. The density was determined by hydrostatic weighing according to Russian Standard GOST 15139-69 (plastics, methods for the determination of density). The liquid medium is distilled water.

Tribological properties and hardness determination ISO 2039-1:2001 were carried out using a CETR UMT-3 friction machine. The pin-on-disk friction scheme was used at a contact pressure of 2 MPa and a sliding speed of 0.25 m/s. The duration of the test is 3 h. The test sample is a cylinder with 10 ± 0.1 mm in diameter and 20 ± 0.1 mm high; the counterbody is a steel disk made of steel No. 45 with a hardness of 45–50 HRC and a roughness R_a_ = 0.06–0.07 µm. The mass of the specimens was measured before and after sliding, using a high precision analytical balance (OHAUS, Parsippany-Troy Hills, NJ, USA). The specific wear rate (SWR) (mm^3^/Nm) was estimated as follows:(1)$$k=\frac{\Delta m}{\rho \times F_{N}\times d^{\prime }}$$
where F_N_, N—normal force; d, m—sliding distance; Δm, g—mass lost during sliding; ρ, g/cm^3^—density of specimens.

Differential scanning calorimetry was performed on a DSC 204 F1 Phoenix (NETZSCH, Selb, Germany), where the measurement error was not more than ±0.1%, the heating rate was 20 °C/min, and the sample weight was 30 ± 1 mg. The measurements were carried out in a helium medium in a temperature range of 40–380 °C. The samples were placed in aluminum crucibles with a 40 µL. Temperature calibration was performed using standard samples of In, Sn, Bi, Pb, and KNO_3_. The enthalpies of fusion were determined by integrating the endothermic peaks between Tm start and Tm end set and used to estimate the degree of crystallinity samples PTFE and PCM by Equation (2):(2)$$\alpha =\frac{\Delta \mathrm{Hm}}{\Delta H\infty }\times 100,$$
where ΔHm is the sample melt enthalpy (J.g^−1^) and Δhe is the theoretical melt enthalpy of a 100% crystalline sample. For PTFE, Δ is value is 82 J.g^−1^ [34,35].

The thermal expansion coefficient was determined according to ISO 11359-2:1999 on a TMA-60/60N series thermomechanical analyzer (Shimadzu, Kyoto, Japan). Measurement of the average coefficient of linear thermal expansion (CLTE) of ᾱ samples with dimensions of 7 × 7 × 2 mm was carried out under a constant load of 4 kPa (~3 g) in the range from 40–200 °C, at a rate of not more than 5° C/min.

IR spectra of PTFE and PCM were obtained on the Varian 7000 FT-IR Fourier transform spectrometer (Varian, Palo Alto, CA, USA) was used to record IR spectra with an attenuated total reflection (ATR) attachment over the range of 550–4000 cm^−1^.

The study of the microstructure and worn surfaces of PTFE and composites based on it was carried out on the JSM-7800F scanning electron microscope (SEM) (JEOL, Tokyo, Japan) in the secondary electron mode at an accelerating voltage of 1–1.5 kV.

## 3. Results

Figure 2 representative stress-strain curves of PTFE and PCM obtained on a universal testing machine. It is known that the introduction of fillers to polymer reduces the elongation at break [24]. At the same time, it can be seen that for composites, the stress increases at small strains for both compression and tensile tests.

Figure 3 shows the results of PTFE and PCM mechanical properties investigated depending on the composition of the fillers. Tensile strength decreases sharply with the introduced of fillers for all composites, in addition, insignificant differences in the tensile strength between PCMs are within the statistical error. However, the decrease in tensile strength is not as strong, in other studies have recorded a stronger decrease in strength relative to the initial polymer [25,36,37,38]. In composites with binary and ternary fillers, the tensile strength decreases by 25–35% compared to the initial PTFE (see Figure 3a). The decrease in the tensile strength properties of PCM can be associated with decrease the effective cross-sectional area that receives the load and formation of defects due to pore formation, and agglomeration of nanoparticles, which reduces the plastic deformation capacity [20,37,38].

Hardness of the PTFE/CF/SiO_2_, PTFE/CF/ZrO_2_, and PTFE/CF/SiO_2_/BN composites increased by ~38% relative to the initial PTFE (Figure 3b). The highest hardness value was achieved for PTFE/CF/ZrO_2_/BN composition, which is 55% higher than the initial polymer. The compressive strength at 5% relative deformation of PTFE/CF/SiO_2_ increased by 30%, and for PTFE/CF/ZrO_2_ by 70% relative to the initial PTFE (Figure 3c). As can be seen in Figure 3d, compressive strength at 10% of relative deformation of PTFE with binary fillers increased by 50–55%, and the compressive strength increased by 69–73% compared to the initial PTFE in composites containing h-BN. Such changes in PCM compressive properties and hardness can be explained by the reinforcing effect of hard carbon fibers [39]. The increase in hardness and compressive strength is due to the fact that part of load is transferred from the polymer matrix to the fibers [40]. The difference in compressive strength between two- and three-component composites is due to the additional content of BN. BN is known to have a platelets shape with a high aspect ratio [25], which additional strengthens the polymer.

The study of PTFE and PCM microstructure was carried out by scanning electron microscopy; the results are shown in Figure 4.

Microstructure of the initial PTFE is characterized by the formation of a lamellae structure, which is typical for this polymer, as shown in Figure 4a. Carbon fibers are visible in all PCM microstructures, which are randomly distributed and chaotically oriented in the polymer volume (Figure 4b–e). The short carbon fibers random orientation in the polymer matrix leads to isotropic material reinforcement, which explains the increase in the compressive strength and composites hardness [41]. Short carbon fibers in the polymer matrix most often form a random distribution, which was also described in the works of other authors [42,43,44]. The SEM method did not reveal the content of nanosized particles in the polymer volume.

Subsequently, thermodynamic properties and density of PTFE and PCM were evaluated, the results are presented in Table 2. Figure 5 illustrates the DSC thermographs of initial PTFE and PCM. Comparison of the melting curves of PTFE and PCM shows that fillers introduced into a polymer matrix shifts the curves of the composites towards an increase in the melting temperature. It can be found that the peaks become larger and the enthalpy of fusion and the degree of crystallinity increase accordingly (Table 2).

Table 2 reveals that introduction of binary (CF + SiO_2_ and CF + ZrO_2_) and ternary fillers into PTFE leads to an increase in the melting point of the polymer from 332.8 to 335.5–336.9 °C. The melting temperature increase in the of composites can be explained by the fact that fillers change the process of PTFE crystallization with the formation of a thermodynamically stable structure of crystals with the chain-extended crystal structure and larger lamellar thickness [45,46]. Therefore, to melt such thick crystals, a higher temperature is required than the melting temperature of the initial polymer. It can be seen that the enthalpy of melting and the degree of crystallinity of PTFE/CF/SiO_2_ increased by 15%, and in the case of PTFE/CF/ZrO_2_ increased by 34% relative to the initial PTFE. The increase of thermodynamic parameters is apparently due to the particles of nanosized filler (SiO_2_ and ZrO_2_) which are additional centers of polymer crystallization. At the same time, decrease in thermodynamic parameters is observed in the PTFE/CF/SiO_2_/BN and PTFE/CF/ZrO_2_/BN composites, which is possibly associated with steric hindrances in the crystallization process due to an increase in the number of crystallization centers, so that the efficiency of additional PTFE crystallization decreases [47].

Among the above characteristics, the CLTE changes strongly, which in composites with binary fillers and PTFE/CF/SiO_2_/BN decreased by 1.5–2.2 times relative to the initial PTFE (Table 2). The lowest CLTE value was demonstrated by the PTFE/CF/ZrO_2_/BN composite, which is ~3.4 times lower than the initial PTFE. A decrease in the CLTE value indicates that composites containing h-BN will be less susceptible to dimensional changes during operation in a friction unit. It should be noted that composites with nanosized ZrO_2_ are characterized by higher values of thermodynamic parameters compared to composites containing nanosized SiO_2_. It has been shown in a number of works that irregularly shaped particles significantly improve the interaction with the polymer matrix [48,49,50]. In this work, it is shown that irregularly shaped ZrO_2_ particles are more actively involved in the crystallization of the polymer matrix, which increases the thermodynamic parameters of PCM.

The introduction of hybrid fillers into PTFE leads to a decrease in density relative to the polymer matrix. This is due to the CF low density (~1.70 g/cm^3^), which is 14.35% lower than the values of the polymer matrix (~2.16 g/cm^3^). In addition, loosening PCM with CF microstructure occurs, which was previously discussed [51]. The density of the composites decreases by 9–11% compared to the initial polymer with the introduction of binary fillers into PTFE. The density increased in composites with ternary fillers, but it is lower than the values of the initial PTFE by 5–7%. It should be noted that polymer composites containing ZrO_2_ are characterized by a higher density, which is consistent with the results of the crystallinity degree obtained by DSC method.

Figure 6 presents the results of PTFE and PCM tribological properties study. The initial PTFE has a low wear resistance and coefficient of friction, which is associated with its weak intermolecular cohesion and the ability to form a thin transfer film on the counterbody [52].

Figure 6 shows that the wear rate in all PCM decreases with the introduction of fillers. The lowest wear rates of 0.53–0.57 × 10^−6^ mm^3^/Nm were revealed in PTFE/CF/SiO_2_ and PTFE/CF/ZrO_2_ composites, which is 845–910 times lower than the unfilled polymer. The specific wear rate is 0.65–0.68 × 10^−6^ mm^3^/Nm with an additional content of nanosized BN (0.5 wt.%) in the PTFE/CF/ZrO_2_/BN and PTFE/CF/SiO_2_/BN composites, which is 708–741 times below the value of the initial polymer matrix. However, it is ~10–20% higher compared to composites containing binary fillers.

The coefficient of friction of PCM with binary fillers in PTFE increased by 18–22% compared to the initial polymer. The coefficient of friction is ~32% higher than the polymer matrix in composites with h-BN. As a rule, solid fillers in the polymer increase the slip resistance of material which leads to a decrease in wear rate and coefficient of friction. A slight increase in the coefficient of friction of PCM is consistent with the known works [53,54]. A slight increase in the coefficient of friction of PCM with three-component fillers compared to two-component fillers may be due to the fact that a larger number of nanoparticles prevent slippage of PTFE chains, increasing the coefficient of friction of the composites [55].

It is known that the tribological properties of material are affected by the processes occurring in the surface layers during sliding [41]. Therefore, the processes occurring on the surface of the material during friction, affect the coefficient of friction and specific wear rate more than its volume properties. Thus, the improvement in the wear resistance of PCM based on PTFE can be explained by a decrease in the propagation of subsurface cracks and a decrease in the size of wear products, as well as the formation of transfer films on the surface of the counterbody. It is known that strong transfer films on contacting surfaces are formed due to the occurrence of tribo-oxidative processes during friction [19,49,56]. Therefore, in order to elucidate the increase in the wear resistance of PCM, studies were carried out using IR spectroscopy (Figure 7). The IR spectrum of the initial PTFE is quite simple and characterized by high peaks between ~1200 cm^−1^ and ~1145 cm^−1^, which are attributed to symmetrical vibrations of the main chain stretching (−CF_2_) [57,58].

As can be seen in Figure 7, the main peaks of the initial PTFE are retained in all samples. The IR spectrum of the friction surface of the initial PTFE is almost identical to the polymer without friction. New peaks appear between 1655 and 1432 cm^−1^ on the worn PCM surfaces. These absorption bands in the IR spectrum indicate the presence of chelated ends of carboxylate polymer chains [59]. The IR spectra of the PTFE/CF/ZrO_2_/BN and PTFE/CF/SiO_2_/BN composites contain small bands at 1377 cm^−1^ related to BN vibrations [60]. The broad band at 3600–3200 cm^−1^ is due to vibrations of (–OH) hydroxyl groups [9]. Similar tribochemical processes during friction of PTFE-based composites indicate the formation of secondary structures [19,51], as well as the formation of a strong transfer film on the metal counterbody [12,54,58], which explains the improvement in the wear resistance of the polymer material. It should be noted that stronger intensity bands are observed in two-component composites compared to composites containing h-BN. It is possible that during friction, h-BN is additionally concentrated on the friction surface, thereby inhibiting oxidation processes. This reduces the formation of secondary structures that additionally protect the material from wear. Thus, the wear resistance of two-component PCMs is somewhat higher compared to composites containing binary fillers. However, this fact has practically no effect on the PCM coefficient of friction, depending on the filler: zirconium dioxide or silicon dioxide.

Figure 8 shows the results of studying the worn surfaces of PTFE and PCM depending on the composition of the fillers by the SEM method. It can be seen that the worn surface of the initial PTFE is quite smooth, which is due to the polymer microstructure and the low cohesion energy density [60]. Studies of worn surfaces PCM show that the fibers are randomly oriented, as well as in the polymer volume.

The fibers are visible on the worn surface of the PCM, where its hardness is much higher than the polymer matrix, thereby protecting the material from its wear (Figure 8). It is known that carbon fibers on the worn PCM surface become microprotrusions, which reduce the contact of polymer with the counterbody [10]. And furthermore, secondary structures in the form of tribofilms are observed between the protruding fibers on the worn surface of the PCM.

Figure 9 shows the worn surface of PTFE/CF/ZrO_2_ at high magnification. It can be seen that the formation of secondary structures on the worn surface protects the PTFE-fiber interfacial layer (indicated by arrows). This is due to the fact that tribofilms perceive part of the frictional load, thereby increasing the actual contact area and reducing the load on individual fibers.

The formation of secondary structures is evidenced by the results of IR spectroscopy for friction surfaces, where oxidative processes occur during friction (Figure 7). Similar tribofilms on a worn surface were found in PTFE filled with carbon fibers and layered silicates [19,51]. The formation of such tribofilms leads to an increase in wear resistance, since the fibers are less pulled out from the worn surface. Thus, the formation of tribofilms on the friction surface of the PCM additionally protects the fibers from destruction, which significantly reduces the material wear.

Figure 10 shows the comparison of tribological properties of the developed PTFE/CF/ZrO_2_ and PTFE/CF/SiO_2_/BN composites with the results of earlier published works [19,61]. PTFE/CF/Zt and PTFE/CF/B (Zt–zeolite, B–bentonite) polymer composites were chosen, where the fiber content is 10 wt.%. and 1 wt.% of natural fillers.

The lowest specific wear rate was shown by the PTFE/CF/ZrO_2_ composite, which is 11% lower than PTFE/CF/Zt and 2 times lower than PTFE/CF/B (Figure 10). For PTFE/CF/SiO_2_/BN composite, the wear resistance is 58% higher compared to PTFE/CF/B, but lower than PTFE/CF/Zt by 10%. The coefficient of friction of polymer composites containing hydrocarbons with natural fillers (bentonite and zeolite) is lower by ~30% compared to PTFE/CF/ZrO_2_ and PTFE/CF/SiO_2_/BN composites. The comparison shows that the modification of PTFE/CF with nanosized zirconia is the most preferred for obtaining the most wear-resistant composite.

## 4. Conclusions

In this work, we have analyzed the mechanical and tribological properties of PCM based on PTFE filled with combined fillers (CF with ZrO_2_, SiO_2_, and h-BN). PCM mechanical testing showed us a slight decrease in tensile strength and an increase in hardness by 38–55% relative to the polymer matrix. It has been shown that introduction of combined fillers into PTFE leads to an increase in compressive strength by 37–73% compared to the polymer matrix. Additional introduction of h-BN into PTFE/CF/ZrO_2_ and PTFE/CF/SiO_2_ leads to an improvement in compressive strength and hardness due to the high aspect ratio of h-BN particles. The study of microstructure of PCM of SEM method showed that the fibers are randomly distributed and oriented in the polymer volume. An assessment of the thermodynamic parameters showed an increase in the temperature and enthalpy of melting, as well as the degree of crystallinity of PCM compared to the initial PTFE. A comparison of two and three-component PCM showed that the introduction of h-BN leads to a decrease in the enthalpy of melting and the degree of crystallinity, but the density is slightly higher. In addition, these values and CLTE for composites with ZrO_2_ are higher than those for SiO_2_. CLTE decrease in all PCM where PTFE/CF/ZrO_2_/BN showed the lowest value, which is 3.4 times lower relatively initial PTFE. The introduction of various fillers combinations significantly increased the PTFE wear resistance. The lowest specific wear rates (0.53–0.57 × 10^−6^ mm^3^/Nm) were shown by PTFE/CF/SiO_2_ and PTFE/CF/ZrO_2_ composites, which is 845–909 times lower than initial PTFE. Wear resistance increased 708–741 times relative to the polymer matrix in composites PTFE/CF/ZrO_2_/BN and PTFE/CF/SiO_2_/BN. A slight decrease in wear resistance and an increase in the friction coefficient of three-component PCM may be associated with a high content of nanosized fillers. IR spectroscopy and SEM studies have shown that the fibers are randomly distributed, and secondary structures in the form of tribofilms are formed on worn surfaces, which are responsible for a significant increase in the wear resistance of composites.

## Figures and Tables

**Figure 1 polymers-15-00313-f001:**
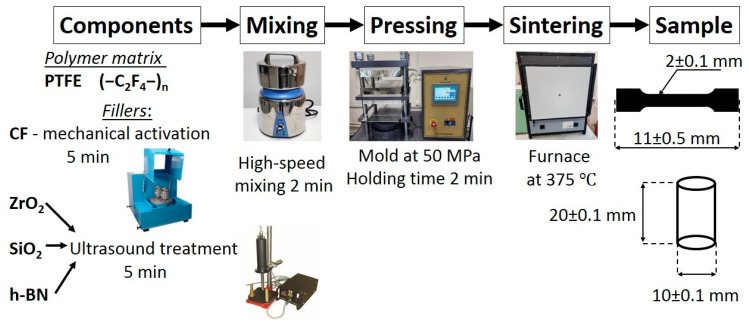
Sample preparation schematically.

**Figure 2 polymers-15-00313-f002:**
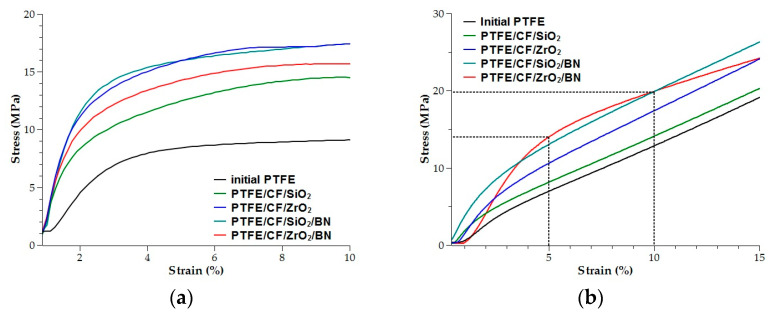
Representative stress-strain curves of PTFE and PCM: (**a**) tensile testing; (**b**) compressive testing.

**Figure 3 polymers-15-00313-f003:**
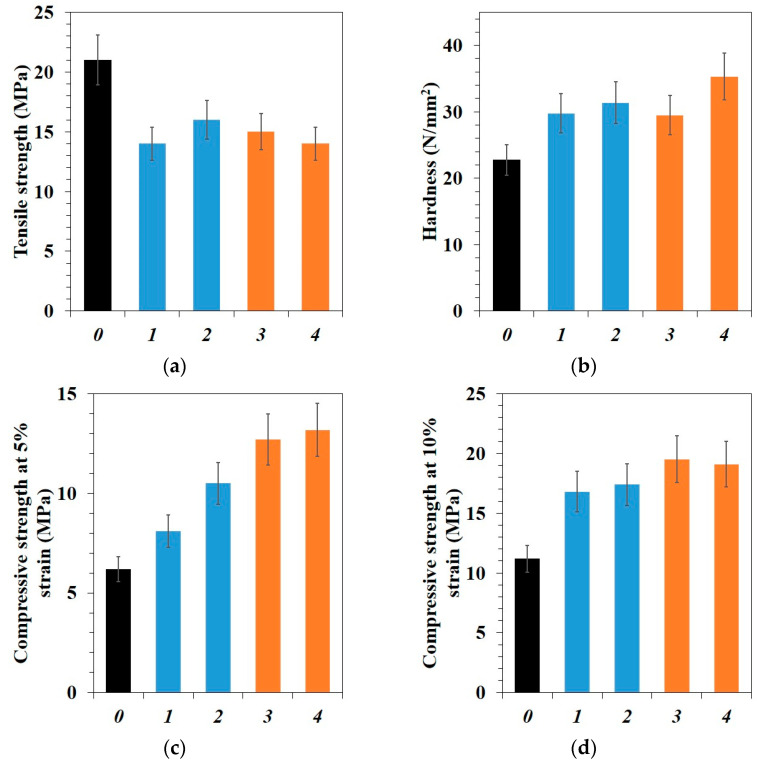
Mechanical properties of initial PTFE and PCM: (**a**) tensile strength; (**b**) hardness; (**c**) compressive strength at 5% relative strain; (**d**) compressive strength at 10% relative strain: *0*–initial PTFE; *1*–PTFE/CF/SiO_2_; *2*–PTFE/CF/ZrO_2_; *3*–PTFE/CF/SiO_2_/BN; *4*–PTFE/CF/ZrO_2_/BN.

**Figure 4 polymers-15-00313-f004:**
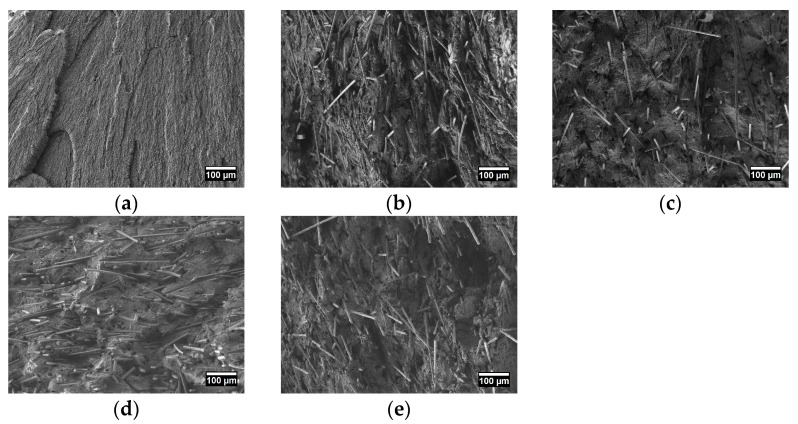
SEM image of microstructures of the initial PTFE and PCM: (**a**) initial PTFE; (**b**) PTFE/CF/SiO_2_; (**c**) PTFE/CF/ZrO_2_; (**d**) PTFE/CF/SiO_2_/BN; (**e**) PTFE/CF/ZrO_2_/BN.

**Figure 5 polymers-15-00313-f005:**
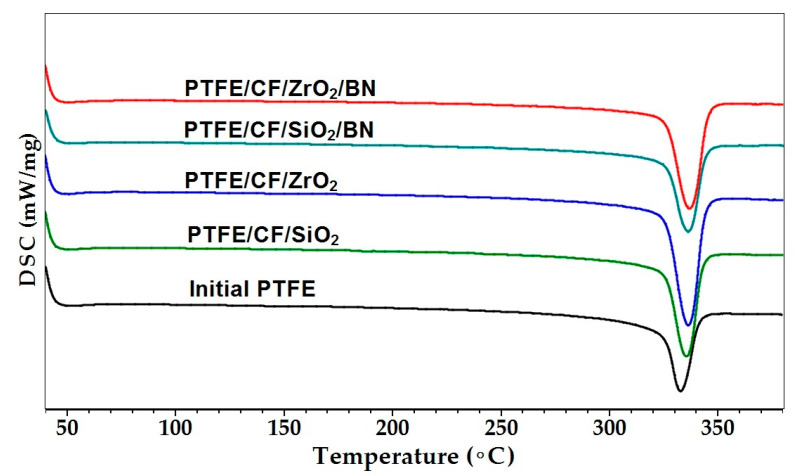
DSC thermographs of the initial PTFE and PCM.

**Figure 6 polymers-15-00313-f006:**
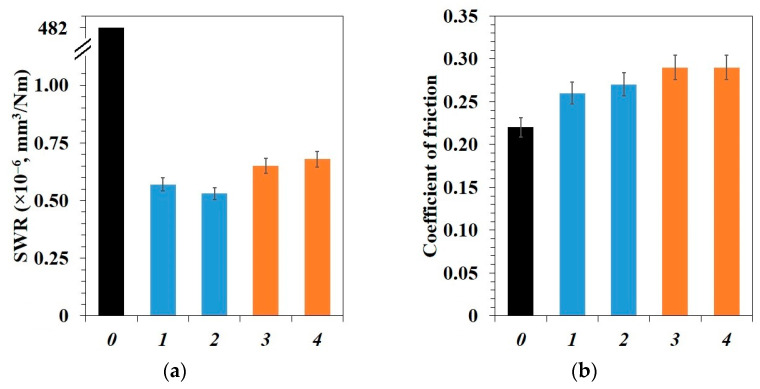
Tribological properties of PTFE and PCM: (**a**) specific wear rate; (**b**) coefficient of friction: *0*–initial PTFE; *1*–PTFE/CF/SiO_2_; *2*–PTFE/CF/ZrO_2_; *3*–PTFE/CF/SiO_2_/BN; *4*–PTFE/CF/ZrO_2_/BN.

**Figure 7 polymers-15-00313-f007:**
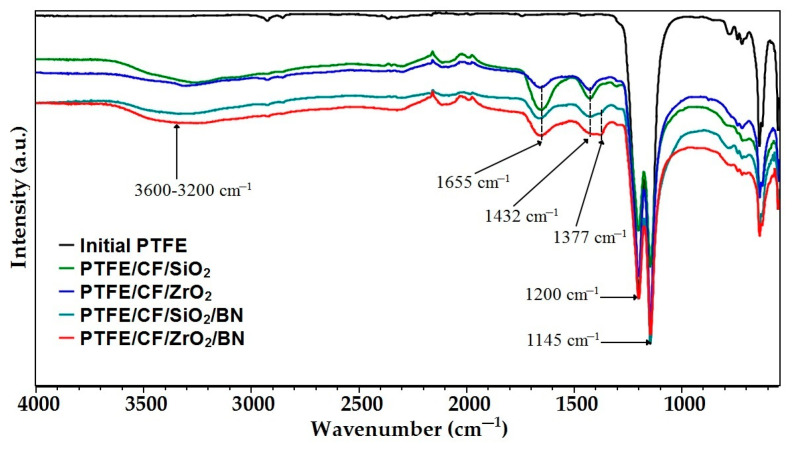
IR spectra of the PTFE and PCM friction surface.

**Figure 8 polymers-15-00313-f008:**
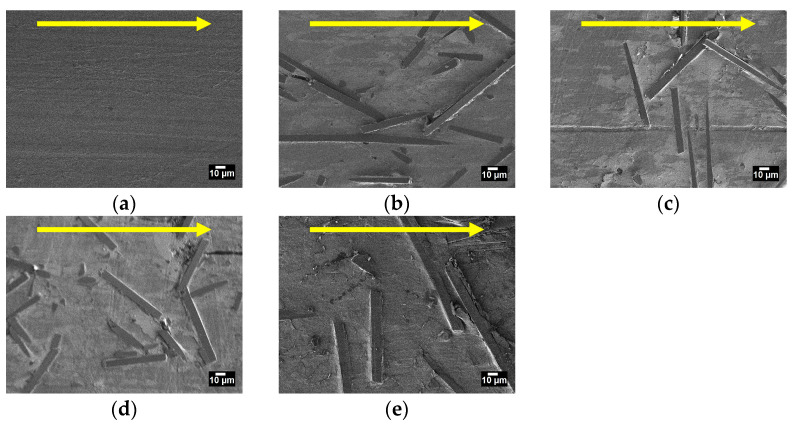
SEM image of worn surfaces of the initial PTFE and PCM: (**a**) initial PTFE; (**b**) PTFE/CF/SiO_2_; (**c**) PTFE/CF/ZrO_2_; (**d**) PTFE/CF/SiO_2_/BN; (**e**) PTFE/CF/ZrO_2_/BN.

**Figure 9 polymers-15-00313-f009:**
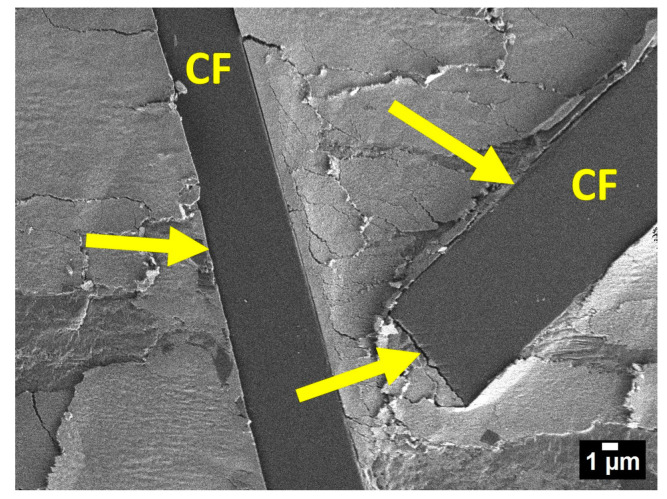
SEM image of worn surface PTFE/CF/ZrO_2_.

**Figure 10 polymers-15-00313-f010:**
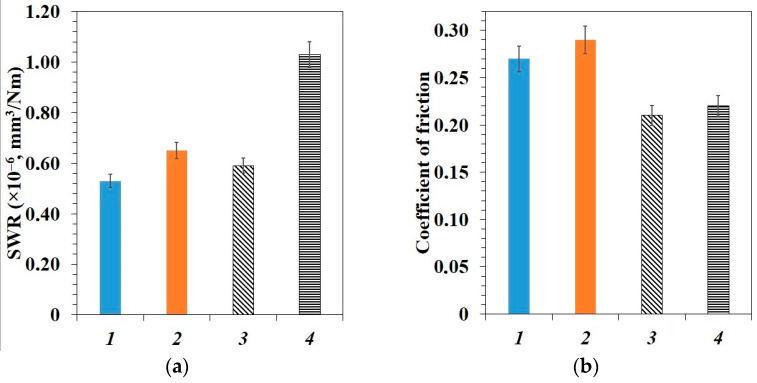
Comparison of PCM specific wear rate (**a**) and coefficient of friction (**b**) developed in this work with previously obtained data: *1*–PTFE/CF/ZrO_2_; *2*–PTFE/CF/SiO_2_/BN; *3*–PTFE/CF/Zt; *4*–PTFE/CF/B.

**Table 1 polymers-15-00313-t001:** Compositions of PTFE composites (wt.%).

Composite	Content of Polymer and Fillers, wt.%				
	PTFE	CF	SiO_2_	ZrO_2_	BN
Initial PTFE	100	-	-	-	-
PTFE/CF/SiO_2_	89	10	1	-	-
PTFE/CF/ZrO_2_	89	10	-	1	-
PTFE/CF/SiO_2_/BN	88.5	10	1	-	0.5
PTFE/CF/ZrO_2_/BN	88.5	10	-	1	0.5

**Table 2 polymers-15-00313-t002:** Thermodynamic properties and density of the initial PTFE and PCM.

Composite	T_m_, °C	ΔH_m_, J/g	α, %	CLTE, 10^−5^, K^−1^	ρ, g/cm^3^
Initial PTFE	332.8	34.3	41.9	26/3	2.16
PTFE/CF/SiO_2_	335.5	39.6	48.3	17.5	1.92
PTFE/CF/ZrO_2_	336.3	45.9	55.9	12.1	1.96
PTFE/CF/SiO_2_/BN	336.3	34.1	41.6	12.6	2.01
PTFE/CF/ZrO_2_/BN	336.9	38.1	46.5	7.8	2.06

## Data Availability

Not applicable.

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
