# Peer review of "Mechanical and Tribological Properties of Polytetrafluoroethylene Modified with Combined Fillers: Carbon Fibers, Zirconium Dioxide, Silicon Dioxide and Boron Nitride"

_polymers, 2023, doi:10.3390/polym15020313_

Round 1
Reviewer 1 Report
Ms. Ref. No.: Polymers-2088173
Title: Mechanical and tribological properties of polytetrafluoroethylene modified with combined fillers: carbon fibers, zirconium dioxide, silicon dioxide and boron nitride
Andrey P. Vasilev et al.
Overall evaluation of the work;
The authors have investigated the effects of ceramic fillers incorporation into carbon-reinforced polymer matrix on the mechanical and tribological properties. The following issues should be addressed to improve the quality and integrity of the manuscript before its publication.
Here are my blind (and some editorial) comments to the authors.
1- The authors mentioned both Russian Standards and ISO standards for the same tests each time, making it unnecessarily long. ISO standards are internationally accepted, so referring to ISO standards would be better to make the manuscript concise.
2- Figure 1a. The authors should explain why the addition of BN nanoflakes decreases the tensile strength of the ZrO2/PTFE composite sample (~17 to 14 MPa), whereas, on the contrary, it increases it for the SiO2/PTFE composite. What statistical analysis test was performed to determine the error bars? Also, there is a typo on Fig. 1a left axis (strenght), which should be fixed. In addition, the figure cap starting with “This Dependence” does not make sense. Should be corrected, and D should be decapitalized.
3- Line 131. The authors say the hardness increases by 38% for all the samples and then claim different values for each sample. Have the authors meant an average increase of 38%?
4- Line 133. Similarly, the authors should explain why adding BN increases the hardness for the ZrO2 composite, whereas it does not seem to have an effect on the SiO2 composite sample (even causes a slight decrease even though staying within experimental error).
5- Figure 2. The fibers seem to align more vertically in Fig 2b and c, whereas in Fig. 2d and e, it is more horizontally. However, on Line 150, the authors claim that there is a random orientation of fibers within the matrix, which leads to isotropic material reinforcement. Has this claim actually been tested to see if the material has isotropic mechanical properties in the other axis or surfaces of the samples?
6- Is there a temperature increase during the friction tests, and if so, how different is the temperature rise/profile between neat PTFE and composite samples? Any differences in the thermal profile of individual ZrO2 and SiO2 samples? If there is a difference, then how would that affect the thermal decomposition or change in the chemical structure of the sample surface (transfer film formation etc.)?
7- Line 143. It would be beneficial to change the word “results” to “micrographs/images” as images are shown in the figure.
8- Line 147. To my knowledge, SEM does not give Supramolecular structure by just imaging, especially at this level of low mag imaging, and supramolecular structure might not be the correct term to use here as the microstructure or morphology is being investigated, and the authors are specifically showing that.
9- Table 2 caption should be rephrased as thermodynamic properties. There is also a note at the bottom of the table saying it may have a footer.
10- There is a typo in Table 1, 0,5 vs. 0.5. Also, the table cap would be more accurate if revised as “ Compositions of PTFE-based composites”
Author Response
Overall evaluation of the work;
The authors have investigated the effects of ceramic fillers incorporation into carbon-reinforced polymer matrix on the mechanical and tribological properties. The following issues should be addressed to improve the quality and integrity of the manuscript before its publication.
Here are my blind (and some editorial) comments to the authors.
- The authors mentioned both Russian Standards and ISO standards for the same tests each time, making it unnecessarily long. ISO standards are internationally accepted, so referring to ISO standards would be better to make the manuscript concise.
Ответ на комментарий №1: Спасибо за комментарий! Исправлено.
2- Рисунок 1а. Авторам следует объяснить, почему добавка нанохлопьев BN снижает предел прочности при растяжении образца композита ZrO 2 /ПТФЭ (~17–14 МПа), а для композита SiO 2 /ПТФЭ, наоборот, повышает ее. Какой тест статистического анализа был проведен для определения планок погрешностей? Также на рис. 1а имеется опечатка по левой оси (сила), которую следует исправить. Кроме того, шапка рисунка, начинающаяся с «Эта зависимость», не имеет смысла. Должен быть исправлен, и D должен быть обезглавлен.
Response to comment #2: The tensile strength of PCM decreases sharply with the introduced of filler content, which means too much filler, especially for nanoparticles, is bad for the tensile strength of PTFE composite, in addition, insignificant differences in the tensile strength between PCMs are within the statistical error. However, other works show a stronger reduction in tensile strength PTFE-composites. The decrease in the tensile strength properties of PCM can be associated with decrease the effective cross-sectional area that receives the load and formation of defects due to a pore formation, and agglomeration of nanoparticles, which reduces the plastic deformation capacity.
3- Line 131. The authors say the hardness increases by 38% for all the samples and then claim different values for each sample. Have the authors meant an average increase of 38%?
Response to comment #3: Thank you for your comment! Hardness increases by ~38% for samples PTFE/CF/SiO2, PTFE/CF/ZrO2, and PTFE/CF/SiO2/BN. The highest hardness value was achieved for the PTFE/CF/ZrO2/BN composition, which is ~55% higher than the initial polymer.
4- Line 133. Similarly, the authors should explain why adding BN increases the hardness for the ZrO2 composite, whereas it does not seem to have an effect on the SiO2 composite sample (even causes a slight decrease even though staying within experimental error).
Response to comment #4: In general, it can be seen that composites with zirconium dioxide are characterized by a higher hardness compared to composites containing ZrO2, although within the statistical error. This is due to changes in the structure of the material, since composites with ZrO2 thermodynamic properties have a higher degree of crystallinity.
5- Figure 2. The fibers seem to align more vertically in Fig 2 b and c, whereas in Fig. 2 d and e, it is more horizontally. However, on Line 150, the authors claim that there is a random orientation of fibers within the matrix, which leads to isotropic material reinforcement. Has this claim actually been tested to see if the material has isotropic mechanical properties in the other axis or surfaces of the samples?
Response to comment #5: The results of mechanical properties testify to the isotropic reinforcement of composites. In tensile testing, the tensile strength is somewhat reduced, but the compressive strength and hardness are increased. Also, on the worn surface, the fibers are oriented in different directions. Short carbon fibers in the bulk of the polymer most often form a random distribution, which was also described in the works of other authors [Lancaster, J. K. (1968). The effect of carbon fibre reinforcement on the friction and wear of polymers. Journal of Physics D: Applied Physics, 1(5), 549. DOI 10.1088/0022-3727/1/5/303; Santos, S. M., Qu, S., & Wang, S. S. (2022). High-Temperature Thermal Transport Properties of Multifunctional PTFE/PEEK-Matrix Composite with Short Carbon Fibers and Graphite Flakes. Journal of Engineering Materials and Technology, 144(4), 041003. https://doi.org/10.1115/1.4054433; Miyase, A., Qu, S., Lo, K. H., & Wang, S. S. (2020). Elevated-temperature thermal expansion of PTFE/PEEK matrix composite with random-oriented short carbon fibers and graphite flakes. Journal of Engineering Materials and Technology, 142(2), 021002. https://doi.org/10.1115/1.4045158; Johansson, P., Marklund, P., Björling, M., & Shi, Y. (2021). Effect of humidity and counterface material on the friction and wear of carbon fiber reinforced PTFE composites. Tribology International, 157, 106869. https://doi.org/10.1016/j.triboint.2021.106869].
6- Is there a temperature increase during the friction tests, and if so, how different is the temperature rise/profile between neat PTFE and composite samples? Any differences in the thermal profile of individual ZrO2 and SiO2 samples? If there is a difference, then how would that affect the thermal decomposition or change in the chemical structure of the sample surface (transfer film formation etc.)?
Response to comment #6: Thank you for your comment! Direct temperature measurement in the PCM friction area was not carried out. In the device during friction, the following temperatures were recorded: for PTFE/CF/ZrO2 ~46 °C and PTFE/CF/SiO2 ~36 °C. Since there is a small change in temperature, this issue was not considered in this work.
7- Line 143. It would be beneficial to change the word “results” to “micrographs/images” as images are shown in the figure.
Response to comment #7: Thank you for your comment! Corrected.
8- Line 147. To my knowledge, SEM does not give Supramolecular structure by just imaging, especially at this level of low mag imaging, and supramolecular structure might not be the correct term to use here as the microstructure or morphology is being investigated, and the authors are specifically showing that.
Response to comment #8: Thank you for your comment! Note Corrected.
9- Table 2 caption should be rephrased as thermodynamic properties. There is also a note at the bottom of the table saying it may have a footer.
Response to comment #9: Thank you for your comment! Note Corrected.
10- There is a typo in Table 1, 0,5 vs. 0.5. Also, the table cap would be more accurate if revised as “Compositions of PTFE-based composites”
Response to comment #10: Thank you for your comment! Corrected.

Reviewer 2 Report
There are not enough reasonable experimental details and sufficient discussion and analysis in this paper. Major revisions are required before this manuscript can be considered for publication.
The authors have demonstrated experimental studies on PTFE polymeric composite materials. However, key experimental details such as characterization of added NPs, formulation schematics, and formulated PCM are missing from the manuscript.This work's novelty must be scientifically addressed. The authors need to address the issue of the appropriate selection of different NPs. Why, for example, is the PCM of solo hBN not included for comparison?
The distribution of NPs and shapes is most crucial for the effective performance of composites, but nothing is mentioned about it. How do you ensure uniform distribution?
In the case of tensile strength, authors have said that "aggregation leads to a drop in tensile strength."
Some parts of the manuscript are also hard to understand because they are not written in a clear way.
1. In the introduction, the authors are advised to include more literature on composite polymer materials. Maybe we can talk about polymer composite bearings with engineered tribo-surfaces.https://doi.org/10.1016/S1572-3364(08)55020-8
Please define "PCM supramolecular structure" in a nutshell.
2. In the entirety of Sections 1 and 2, very few references are cited. Therefore, the author should find reliable documents from literature for reference and include them at the appropriate places.
3. Authors need to corroborate their results for enhanced hardness with published literature ("hhighest value of compressive strength at 105%").
136 and 112% compared to the initial PTFE were achieved for PTFE/CF/SiO2/BN and
137 PTFE/CF/ZrO2/BN. Why is this the case?
4. In context to "the PCM compressive properties and hardness can be explained by 141, the reinforcing effect of hard carbon fibers," why different values of all PCM when there is the same 10% of CNF in each?
It would be fantastic if, in addition to SEM images, EDX mapping could be included to justify the dissection of accurately characterised CNF fibres and NPs.
How does the author calculate the thermodynamic properties of the PCM and make the case for RM elaboration?
What does the author mean here in line 162: "with an elongated chain and a thickening of the lamellar layer of..."
Please justify how "which is possibly associated with steric hindrances" comes into play for a solid PCM system? I think they should be accountable initially during formulation, unlike liquid suspensions such as emulsions and nanoparticle-doped lubricants.
line "about 172-179 about CLTE" refers to which results?
The SWR of the specific wear rate of the initial PTFE is mm3/Nm.
why it is not included in BAR as a reference for comparison. and why is PTFE+CNF not regarded as a reference point for compassion?
Please explain why (845-910) times lower should be PTFE-CNF as.53-0.57 10-6 mm3/Nm for PTFE/CF/SiO2 199 and PTFE/CF/ZrO2 composites, whereasPlease correct and justify.
The friction increases with the addition of NPs. Please refer to more literature and justify. How does the combination of NPs cause antogostic friction?please elaboarte.
Explain the polar opposite behaviour of friction and SWR.whereas many researchers have yielded enhanced PV values of PCM along with decreased friction with the addition of SLs nanofiller for PAEK and PEAK PCM. What could be causing this behavior?
How come an author jumps to "irregular"shape of ZrO2 breaks the polymer chains compared to particles with a spherical shape of SiO2," when the shape of NPs is not disclosed in the manuscript?
Please justify the statement that "oxidation processes are accelerated and deepened during the friction 235 of the PCM, thereby the transfer films are formed earlier." No evidence is provided.
Which formation of secondary structures on the worn surface protects the PTFE-255 fibre interfacial layer?
No clear evidence is provided.
How does the addition of SiO2, ZrO2, or ceramic particles affect wear via hardened tribofilm formation? Authors must provide solid evidence of improved tribo performance as well as reasoning from the literature.

Author Response
There are not enough reasonable experimental details and sufficient discussion and analysis in this paper. Major revisions are required before this manuscript can be considered for publication.
The authors have demonstrated experimental studies on PTFE polymeric composite materials. However, key experimental details such as characterization of added NPs, formulation schematics, and formulated PCM are missing from the manuscript. This work's novelty must be scientifically addressed. The authors need to address the issue of the appropriate selection of different NPs. Why, for example, is the PCM of solo hBN not included for comparison?
The distribution of NPs and shapes is most crucial for the effective performance of composites, but nothing is mentioned about it. How do you ensure uniform distribution?
In the case of tensile strength, authors have said that "aggregation leads to a drop in tensile strength."
Some parts of the manuscript are also hard to understand because they are not written in a clear way.
Response to comment: Uniform distribution of filler particles is achieved by mechanical activation. The introduction section has been corrected (relevance).
- In the introduction, the authors are advised to include more literature on composite polymer materials. Maybe we can talk about polymer composite bearings with engineered tribo-surfaces.https://doi.org/10.1016/S1572-3364(08)55020-8
Please define "PCM supramolecular structure" in a nutshell.
Response to comment #1: Thank you for your comment! Corrected.
- In the entirety of Sections 1 and 2, very few references are cited. Therefore, the author should find reliable documents from literature for reference and include them at the appropriate places.
Response to comment #2: Thank you for your comment! Note corrected.
- Authors need to corroborate their results for enhanced hardness with published literature ("highest value of compressive strength at 105%").
136 and 112% compared to the initial PTFE were achieved for PTFE/CF/SiO2/BN and
137 PTFE/CF/ZrO2/BN. Why is this the case?
Response to comment #3: Thank you for your comment! Corrected. The increase in hardness and compressive strength is due to the fact that part of load is transferred from the polymer matrix to the fibers.
- In context to "the PCM compressive properties and hardness can be explained by 141, the reinforcing effect of hard carbon fibers," why different values of all PCM when there is the same 10% of CNF in each?
Response to comment #4: Obviously, this is already associated with structural changes in composites upon the introduction of nanosized fillers. In general, it can be seen that composites with zirconium dioxide are characterized by a higher hardness compared to composites containing ZrO2. This is due to a change in the structure of the material, since composites with the thermodynamic properties of ZrO2 have a higher degree of crystallinity compared to composites containing SiO2.
It would be fantastic if, in addition to SEM images, EDX mapping could be included to justify the dissection of accurately characterised CNF fibres and NPs.
Response to comment: Now we cannot make an EDX, in the future we will definitely try to find such a device.
How does the author calculate the thermodynamic properties of the PCM and make the case for RM elaboration?
Response to comment: Thermodynamic properties obtained on a DSC 204 F1 Phoenix (NETZSCH, Germany). The enthalpies of fusion were determined by integrating the endothermic peaks between Tm start and Tm endset and used to estimate the degree of crystallinity samples PTFE and PCM by Equation (2):
(4)
where ΔHm is the sample melt enthalpy (J.g−1) and ΔH∞ is the theoretical melt enthalpy of a 100% crystalline sample. For PTFE, ΔH∞ value is 82 J.g−1 [10.1590/1980-5373-mr-2017-0326 https://doi.org/10.1016/j.polymdegradstab.2019.109053].
What does the author mean here in line 162: "with an elongated chain and a thickening of the lamellar layer of..."
Response to comment: Corrected. We meant the following: …the chain-extended crystal structure and larger lamellar thickness…
Please justify how "which is possibly associated with steric hindrances" comes into play for a solid PCM system? I think they should be accountable initially during formulation, unlike liquid suspensions such as emulsions and nanoparticle-doped lubricants.
line "about 172-179 about CLTE" refers to which results?
Response to comment: steric hindrances in the crystallization process due to an increase in the number of crystallization centers
Response to comment: Thank you for your comment! Corrected. CLTE refers to Table 2.
The SWR of the specific wear rate of the initial PTFE is mm3/Nm.
Response to comment: Corrected.
why it is not included in BAR as a reference for comparison. and why is PTFE+CNF not regarded as a reference point for compassion?
Response to comment: Note corrected. As a comparison, we took the initial (neat) PTFE
Please explain why (845-910) times lower should be PTFE-CNF as.53-0.57 10-6 mm3/Nm for PTFE/CF/SiO2 199 and PTFE/CF/ZrO2 composites, whereas. Please correct and justify.
Response to comment: As a comparison, we took the initial (neat) PTFE
The friction increases with the addition of NPs. Please refer to more literature and justify. How does the combination of NPs cause antogostic friction?please elaboarte.
Response to comment: With a higher content of NP during friction, the friction moment increased, which also increased the coefficient of friction. It is also possible that higher NP composites form a thicker transfer film during adhesive wear.
Explain the polar opposite behavior of friction and SWR.whereas many researchers have yielded enhanced PV values of PCM along with decreased friction with the addition of SLs nanofiller for PAEK and PEAK PCM. What could be causing this behavior?
Response to comment: Due to the large content of NP, i.e. up to a certain content, the PTFE matrix will not be able to bond well with the NPs.
How come an author jumps to "irregular" shape of ZrO2 breaks the polymer chains compared to particles with a spherical shape of SiO2," when the shape of NPs is not disclosed in the manuscript?
Response to comment: Note corrected.
Please justify the statement that "oxidation processes are accelerated and deepened during the friction 235 of the PCM, thereby the transfer films are formed earlier." No evidence is provided.
Response to comment: Corrected.
Which formation of secondary structures on the worn surface protects the PTFE-255 fibre interfacial layer?
No clear evidence is provided.
Response to comment # In previous studies, it was shown that the introduction of layered silicates with carbon fibers also leads to the formation of secondary structures in the form of tribofilms, while it is not detected for PTFE + CF. In this work, tribofilms were detected by SEM and IR spectroscopy; they are visually distinguishable by microscopy and are detected by IR spectroscopy as oxidized forms.
How does the addition of SiO2, ZrO2, or ceramic particles affect wear via hardened tribofilm formation? Authors must provide solid evidence of improved tribo performance as well as reasoning from the literature.
Response to comment Thank you for your comment! Note corrected.

Reviewer 3 Report
1. What is the basis of doing Ultrasonication for 5 minutes? Is any reference time considered based on the literature?
2. Similar question follows for the Mechanical Activation using a planetary mill?
3. The wt% for mixing in PTFE is not equal wrt ZrO2 and BN? What is the rationale behind fixing the wt.%?
4. I suggest re-writing the paragraph from Line 103 till Line 113. The sentence formation is a bit confusing to the readers.
Author Response
1. What is the basis of doing Ultrasonication for 5 minutes? Is any reference time considered based on the literature?
Response to comment #1: Note corrected.
2. Similar question follows for the Mechanical Activation using a planetary mill?
Response to comment #2: Thank you for your comment! Corrected.
3. The wt% for mixing in PTFE is not equal wrt ZrO2 and BN? What is the rationale behind fixing the wt.%?
Response to comment #3: In previous works, we have studied the effect of nanosized particles at different contents. We realized that adding more than 0.5 BN leads to a significant decrease in mechanical properties. Therefore, we chose such ratios of fillers in this work.
4. I suggest re-writing the paragraph from Line 103 till Line 113. The sentence formation is a bit confusing to the readers.
Response to comment #4: Thank you for your comment! Corrected.

Reviewer 4 Report
Dear Authors
The manuscript could be published after considering following major revision:
1-Represent some original results of various tests (mechanical, DSC, etc.) as sample. Final results representation in one chart is not enough.
2-It is better to represent sample preparation schematically. It helps readers quickly understanding the procedure.
2- Highlight the importance, necessity and advantage of your study.
3-Correct grammatical and spelling errors.
Sincerely
Author Response
Dear Authors
The manuscript could be published after considering following major revision:
1-Represent some original results of various tests (mechanical, DSC, etc.) as sample. Final results representation in one chart is not enough.
Response to comment #1: Thank you for your comment! Note corrected.
2-It is better to represent sample preparation schematically. It helps readers quickly understanding the procedure.
Response to comment #2: Note corrected. Added technological scheme of processing.
3- Highlight the importance, necessity and advantage of your study.
Response to comment #3: The relevance of the work has been corrected.
4-Correct grammatical and spelling errors.
Response to comment #4: Thanks for the note, fixed.

Round 2
Reviewer 1 Report
The authors have tried to address my comments. I recommend the acceptance for publication.
Author Response
Thank you very much! And Happy New Year!
Reviewer 2 Report
the Author has incorporated some of the queries. However still, the manuscript is ambiguous at various locations as highlighted in PDF. The authors have failed to mark a clear demarcation difference between; the addition of 2 fillers composite and Three.
The role of hbN is a composite is not discussed; in fact, it lead to an antagonistic effect; The author needs to add this in the conclusion & abstract.
Many more Queries are also commented on inside highlighted PDF. Like Lubrication mechanism & oxidation as acclaimed by authors.
Authors are suggested to properly address all Queries and incorporated in articles
Author Response
Response to the Reviewers’ Comments Round 2
Reviewer #2
the Author has incorporated some of the queries. However still, the manuscript is ambiguous at various locations as highlighted in PDF. The authors have failed to mark a clear demarcation difference between; the addition of 2 fillers composite and Three.
The role of hbN is a composite is not discussed; in fact, it lead to an antagonistic effect; The author needs to add this in the conclusion & abstract.
Many more Queries are also commented on inside highlighted PDF. Like Lubrication mechanism & oxidation as acclaimed by authors.
Authors are suggested to properly address all Queries and incorporated in articles
Answer:
Dear Reviewer, we have tried to answer all your questions. Merry Christmas and Happy New Year 2023!

Reviewer 3 Report
I am satisfied with the review changes which i suggested the authors to incorporate. The paper can be accepted for publication. Good luck to the authors
Author Response
Thank you very much!
Happy New Year!
Reviewer 4 Report
Dear Authors
for the following comment:
"Represent some original results of various tests (mechanical, DSC, etc.) as sample. Final results representation in one chart is not enough"
1-If I understand correctly, you just represent original mechanical test results in the reply paper (the graphs did not introduce). The titles of graph axes must be in English. The quality of graph pictures must be increased.
2-Origininal DSC test results are not represented.
So, representation of some original results of various tests (mechanical, DSC, etc.) as sample must be added in the paper. The added original graphs must be introduced with a clear title. The titles of graph axes must be in English
After major correction the manuscript could be published.
Sincerely
Author Response
Response to the Reviewers’ Comments Round 2
Reviewer #4
for the following comment:
"Represent some original results of various tests (mechanical, DSC, etc.) as sample. Final results representation in one chart is not enough"
1-If I understand correctly, you just represent original mechanical test results in the reply paper (the graphs did not introduce). The titles of graph axes must be in English. The quality of graph pictures must be increased.
2-Origininal DSC test results are not represented.
So, representation of some original results of various tests (mechanical, DSC, etc.) as sample must be added in the paper. The added original graphs must be introduced with a clear title. The titles of graph axes must be in English
After major correction the manuscript could be published.
Answer:
Dear Reviewer, we have tried to answer all your questions. Merry Christmas and Happy New Year 2023!

Round 3
Reviewer 2 Report
The authors have incorporated all of the queries.
Author Response
Thank you very much!
Reviewer 4 Report
Dear Authors
Some comments are considered and the paper was corrected. But,
Original mechanical test graphs were not included into the paper, “with the manner and requirements that mentioned in the previous comments”.
Please pay attention to the comments, completely.
After adding the wanted requirement, the manuscript could be published.
Sincerely
Author Response
Ответ на комментарии рецензентов, раунд 3
Рецензент №4
Некоторые замечания учтены, статья исправлена. Но,
Оригинальные графики механических испытаний не были включены в документ «в порядке и с требованиями, указанными в предыдущих комментариях».
Пожалуйста, обратите внимание на комментарии, полностью .
После добавления требуемого требования рукопись может быть опубликована.
Отвечать:
Уважаемый рецензент, мы включили в статью репрезентативные оригинальные тестовые кривые.
